# Physiotherapy in the Recovery of Paraplegic Dogs without Nociception Due to Thoracolumbar Intervertebral Disc Extrusion Treated Surgically

**DOI:** 10.3390/ani14182648

**Published:** 2024-09-12

**Authors:** Júlia da Silva Rauber, Julya Nathalya Felix Chaves, Mathias Reginatto Wrzesinski, Amanda Miwa Takamori Sekita, Thais da Silva Soares, Diego Vilibaldo Beckmann, Alexandre Mazzanti

**Affiliations:** 1Graduate Program in Veterinary Medicine, Veterinary Neurology and Neurosurgery Service, Federal University of Santa Maria, Center for Rural Sciences, University Veterinary Hospital, Santa Maria 97105-900, RS, Brazil; julia.dsrauber@gmail.com (J.d.S.R.); julyachavesb@hotmail.com (J.N.F.C.); mathias_reginatto@hotmail.com (M.R.W.); miwa.sekita@acad.ufsm.br (A.M.T.S.); thais.soares@acad.ufsm.br (T.d.S.S.); 2Department of Small Animal Clinic, Veterinary Neurology and Neurosurgery Service, Federal University of Santa Maria, Center for Rural Sciences, University Veterinary Hospital, Santa Maria 97105-900, RS, Brazil; beckmann.diego@ufsm.br

**Keywords:** IVDE, physiotherapy, ambulation, surgery, dog, neuroreabilitation

## Abstract

**Simple Summary:**

This study addresses the effects of postoperative physiotherapy in dogs with intervertebral disc extrusion treated surgically. The animals were divided into two groups: the physiotherapy group (PG), which included those that underwent decompressive surgery and postoperative physiotherapy; and the control group (CG), which included dogs that did not undergo any physiotherapy after surgery. The physiotherapy protocol began immediately after surgery. A total of 51 dogs were included, with 30 in the PG and 21 in the CG. The functional recovery rate in dogs up to 21 days postoperatively was 10% (3/30) in the PG and 19% (4/21) in the CG. After 21 days postoperatively, the rates were 43.33% (13/30) in the PG and 61.9% (13/21) in the CG, with no observed difference between the groups (*p* = 0.258). Physiotherapy administered twice a week in paraplegic dogs with loss of nociception due to thoracolumbar intervertebral disc extrusion does not seem to influence functional recovery compared to the group without physiotherapy.

**Abstract:**

Several authors have advocated for the role of physiotherapy in canine intervertebral disc extrusion, and it is routinely recommended by various veterinary neurologists. However, veterinary literature does not unanimously support the routine use of physiotherapy to ensure an increase in locomotor return in dogs with IVDE. The aim of the study was to investigate whether physiotherapy can influence the functional recovery of paraplegic dogs with loss of nociception (LN) affected by thoracolumbar IVDE (Hansen type I) and treated surgically. The animals were divided into two groups: the physiotherapy group (PG), which included those that underwent decompressive surgery and postoperative physiotherapy; and the control group (CG), which included dogs that did not undergo any physiotherapy after surgery. A total of 51 dogs were included, with 30 in the PG and 21 in the CG. The number of physiotherapy sessions ranged from 6 to 60. The rate of functional recovery in dogs within 21 days postoperatively (PO) was 10% (3/30) in the PG and 19% (4/21) in the CG. After 21 days PO, the recovery rate was 43.33% (13/30) in the PG and 61.9% (13/21) in the CG, with no significant difference observed between the groups (*p* = 0.258). Based on the findings of this study, it was concluded that physiotherapy in paraplegic dogs with LN due to thoracolumbar IVDE does not appear to influence functional recovery compared to the group without physiotherapy.

## 1. Introduction

Intervertebral disc extrusion (IVDE) (Hansen type I) is recognized as one of the main causes of myelopathy in dogs [1]. The occurrence of this disease is characterized by the degeneration, rupture, and protrusion of the intervertebral disc into the vertebral canal, resulting in the contusion and compression of the spinal cord, which can lead to loss of motor and sensory function in the pelvic limbs [2].

Since the 1990s, there has been a notable increase in the application of physiotherapy techniques in animals, particularly due to the growing expectations of pet owners and technological advancements in this field. The results observed in experimental studies of spinal cord injuries and clinical trials involving human patients have drawn the attention of veterinarians, prompting a reevaluation of postoperative management of their patients [3]. Currently, the neurological caseload of veterinary rehabilitation centers predominantly consists of dogs with spinal cord conditions [4].

Several authors have advocated for the role of physiotherapy in canine IVDE and it is routinely recommended by numerous veterinary neurologists [5,6,7]. However, the current state of veterinary literature does not unanimously support the role of routine physiotherapy in ensuring improved locomotor recovery in dogs with IVDE [8]. These same authors proposed investigating the impact of physiotherapy in dogs with severe spinal cord injuries, suggesting that postoperative physiotherapy in this cohort might result in higher recovery rates.

The objective of this retrospective study was to investigate whether postoperative physiotherapy can assist in the functional neurological recovery of paraplegic dogs with loss of nociception (LN) affected by thoracolumbar IVDE and treated surgically.

## 2. Materials and Methods

The study selected dogs treated at the Veterinary Neurology and Neurosurgery Service (VNNS) of an educational institution from November 2021 to January 2024, with a confirmed diagnosis of thoracolumbar IVDE.

### 2.1. Participants

Only paraplegic dogs with a history of LN within 96 h prior to presentation and that underwent decompressive surgery, with or without postoperative physiotherapy, were included.

It was assumed that the LN occurred concurrently with the total loss of pelvic limb movement reported by the owners [9,10,11,12] and persisted until the date of surgery. All surgeries were performed within 24 h of the dog’s admission to the hospital.

The presence or absence of nociception was assessed by applying strong pressure to the nail base, digits of the pelvic limbs, and tail base using hemostatic forceps [11,13,14,15,16,17]. The absence of a detectable behavioral response (crying, licking, attempting to bite, or turning the head toward the stimulus) or a physiological response (pupil dilation, increased heart rate, or respiratory rate) to repeated stimuli was considered indicative of LN [10,11,15,16,18,19]. The presence or absence of nociception was determined by the veterinarians at the VNNS.

To confirm IVDE, dogs underwent imaging exams (myelography or computed tomography) to determine the location and side of spinal cord compression. The surgical technique employed for spinal cord decompression was dorsolateral hemilaminectomy associated with fenestration of the affected intervertebral disc [20].

Decompressive surgery was performed as soon as possible after diagnosis. Following surgery, dogs from both the physiotherapy group (PG) and the control group (CG) remained hospitalized for three days. All dogs were treated with manual bladder compression three times a day [21]. Additionally, they received analgesia with methadone (0.3 mg/kg, six times a day) and dipyrone (25 mg/kg, three times a day). In the immediate postoperative period, all dogs received a single dose of dexamethasone (0.25 mg/kg) intravenously.

### 2.2. Study Design

The dogs were divided into two groups: the PG, which included dogs that underwent decompressive surgery and physiotherapy starting immediately postoperatively and continued for several weeks, and the CG which included dogs that underwent only decompressive surgery. After hospital discharge, the dogs remained indoors without engaging in walks or more intense exercises.

### 2.3. Inclusion Criteria

The inclusion criteria for the PG and CG groups required complete neurological clinical records, including breed, age, sex, history detailing the onset and duration of clinical signs, the results of physical and neurological exams obtained during consultation, definitive diagnosis of IVDE via surgical procedure and removal of disc material from the vertebral canal, degree of neurological dysfunction before and after surgery, location of compression, and absence of orthopedic comorbidities.

In addition to the aforementioned criteria, dogs in the PG included those with complete physiotherapy clinical records, detailing the degree of neurological dysfunction before starting physiotherapy protocols and data regarding the employed protocol. For inclusion in the CG, the absence of postoperative physiotherapy had to be documented in the neurological records. To confirm this information, pet owners were contacted via telephone to ensure that no physiotherapy modalities were performed at home or in specialized centers.

### 2.4. Physiotherapy Protocol

All animals in the PG began physiotherapy treatment immediately postoperatively with cryotherapy. This modality was applied for the first 72 h after surgery, using crushed ice wrapped in a plastic bag, applied over the surgical incision for 20 min, four times a day [22]. Other physiotherapy protocols were conducted in one-hour sessions, twice a week.

The introduction of each physiotherapy technique was determined based on the individual progression of each patient. These techniques were grouped into protocols, which progressed according to the patient’s evolution. The initial protocol (P1) consisted of a kneading massage (pétrissage technique), passive range of motion (PROM), flexor reflex stimulation, and neuromuscular electrical stimulation (NMES). During the application of these modalities, the animals were positioned in lateral recumbency on a mattress, and after each procedure performed on one pelvic limb, the same sequence was repeated on the contralateral limb.

The massage was performed using the deep kneading technique (pétrissage). This was carried out to warm up the muscles, allowing for better performance during passive joint movement exercises. The thumbs were placed on one side of the muscle and the fingers on the other, using both hands. With opening and closing hand movements, the thigh muscles were kneaded for five minutes. The increase in pressure on the muscles was gradual and according to the patient’s tolerance. Then, the extension and flexion stretches of each joint of the affected limb and before PROM exercises. For passive stretching, the bones proximal and distal to the joint being stretched were stabilized with hands, and an increase in the range of motion in flexion was initiated until a restriction to the movement was obtained and tolerated by the animal. The stretch was held for 20 s. After the flexion stretch, the pressure was slowly released, and the extension stretch of the same duration was started [23].

To perform the passive joint movement, one hand was placed on the part of the limb above the joint, and the other hand on a part of the limb below the joint being moved. Once the hands were in the correct position and the limb was supported, the joint was flexed to the maximum amplitude tolerated by the animal. Then, with the hands maintained in the same positions, the joint was slowly extended to the limit tolerated by the patient. This was performed with 30 cycles (one cycle corresponding to one extension and one flexion) individually on each joint of the pelvic limbs, simultaneously with a bicycle motion to simulate ambulation [23]. After the PROM, the flexor reflex stimulation was repeated 15 times [24].

To perform the flexor reflex exercise, a manual stimulus (pinching) was applied to the interdigital skin of the pelvic limb. As the limb flexed, resistance was achieved by holding the foot, creating a gentle “tug-of-war” where the patient pulled more forcefully to withdraw the limb from the therapist’s grip. Five repetitions were performed per limb [24].

Medium-frequency NMES (Russian stimulation) was applied to the pelvic limbs (vastus lateralis and biceps femoris) for 15 min using specific parameters, including a current frequency of 2500 Hz, a pulse width of 50%, stimulation cycles of 12 s followed by 25 s of rest (on ratio of 1:2), and three-second rise and decay ramps in reciprocal mode. Electrodes were positioned quadripolarly over the thigh muscles, with the current intensity adjusted according to the patient’s tolerance [4].

Starting from P1, the physiotherapy modalities were adapted or replaced based on each patient’s progress and adaptation. For dogs that regained movement in their pelvic limbs, a second protocol (P2) was added. This included walking on an underwater treadmill at a minimum speed of 1.5 km/h, with the water level at the greater trochanter, gradually increasing the activity time from 5 to 15 min [2]. They were guided by a veterinarian using a chest harness and leash and motivated with treats and owner stimulation. The animal’s weight was supported by a body sling. As the animal improved its weight-bearing ability, the sling was adjusted to provide less support until the animal could sustain its own weight. If the animal showed signs of fatigue, the exercise was stopped. The maximum duration for this activity was 15 min. Assisted walks were also part of this protocol and were conducted under the supervision of a veterinarian. The animal was supported by a body sling and encouraged to walk on different surfaces, such as non-slip flooring, concrete, and grass. Walks lasted three to five minutes at the pace the animal was comfortable with [4]. Higher or lower speeds were not encouraged. If the dog showed signs of fatigue, the exercise was stopped and resumed only in the next session.

When the animals could take a few steps, active therapeutic exercises (P3) were introduced, which included walking on a mattress and walking over obstacles, as well as using a circular proprioceptive platform. With a chest harness and leash, the animal was encouraged to walk on a mattress. The body support was gradually reduced according to the patient’s progress. Treats were used as motivation. The walks were slow and if the animal showed signs of fatigue or reluctance to continue the exercise, the session was stopped. With a harness and leash, the animal was encouraged to walk over obstacles such as bars and through paths marked by cones. Obstacle walks involved five obstacles at a height of six centimeters from the ground, with the distance between them adjusted to the animal’s body length, repeated 5 to 10 times, along with walks on a three-meter-long mattress [4]. For dogs that could maintain weight-bearing in a standing position for at least five seconds, a circular proprioceptive platform was added to the protocol for two to three minutes. During this activity, the animal stood with all four limbs on the platform, allowing for 360° movements. The animals stood on the platform for two to three minutes [25,26].

### 2.5. Evaluation Postoperative

Functional neurological recovery was classified as satisfactory when achieved complete neurological recovery; that is, the animal could take 10 steps without falling, unassisted, and with nociception present [7,13]. Dogs that did not regain the ability to walk or showed no change in their initial clinical status were defined as having an unsatisfactory recovery [15]. Dogs that regained the ability to walk but did not recover nociception (spinal walking) were also considered to have had an unsatisfactory recovery. In this article, successful recovery is defined as both sensory and motor recovery [15]. The maximum follow-up time for dogs after surgery was 120 days. Dogs that regained the ability to walk but did not recover nociception (spinal walking) were also classified as having an unsatisfactory recovery. In this study, a successful recovery is defined as achieving both sensory and motor function restoration. The maximum follow-up period for dogs post-surgery was 120 days.

### 2.6. Statistical Analysis

Functional recovery was evaluated in two periods based on the time taken to achieve it: up to 21 days and more than 21 days post-surgery. The correlation between functional recovery and the duration of deep pain perception (DPP) loss in the PG and CG groups was assessed using the Chi-square test, and the correlation of spinal walking occurrence between the groups was evaluated using the Chi-square test. Statistical analyses were performed using the Jamovi software (version 1.6.23), with significance set at *p* < 0.05.

## 3. Results

The total sample of this study was 51 dogs, distributed into PG (*n* = 30) and CG (*n* = 21). Acknowledging that the size of PG and CG could impair the analysis, Cohen’s d tests were performed, when applicable, and the effect size was always small.

The results concerning breed, age, sex, compression site, duration of nociception loss, functional neurological recovery, and time functional recovery of both groups are described in Table 1, of the GC only in Table 2 and of PG in Table 3. The most affected breed in the studied population was the Dachshund, accounting for 66.7% (34/51). The age of the dogs ranged from 2 to 10 years, with an average of 5.5 ± 1.8 years. Regarding sex, 50.9% (26/51) were female and 49.1% (25/51) were male. The compression site varied between T11–T12 and L2–L3, with the T13–L1 space being the most affected, at 49.1% (25/51).

In the dogs from PG, the median number of sessions performed was 10 ± 5.07, and the median duration of physiotherapy treatment was 8.85 ± 5.12 weeks. Five dogs (16.7%) underwent only the modalities in protocol 1 (P1), ten dogs (33.3%) underwent both protocol 1 and protocol 2 (P2), and fifteen dogs (50%) completed all protocols (Table 3).

In terms of dogs achieving satisfactory recovery within 21 days post-operation, 23.1% (3/13) of dogs in PG and 31% (4/13) in CG reached this condition within the same period. Among dogs achieving satisfactory recovery after 21 days, it was observed that 76.9% (10/13) were in PG and 69% (9/13) in CG (Figure 1). There was no statistically significant difference between these results in the two groups (*p* = 1.000).

Regarding functional neurological recovery, 13 out of 30 dogs (43.3%) in PG achieved satisfactory recovery compared to 13 out of 21 dogs (61.9%) in CG. Unsatisfactory recovery was observed in 17 out of 30 dogs (56.7%) in PG and 8 out of 21 dogs (38.1%) in CG (Figure 2). There was no statistically significant difference observed in comparing these findings (*p* = 0.258).

Regarding the return of spinal walking without the recovery of nociception, it was observed that 22.6% (7/30) of dogs in PG and 14.3% (3/21) in CG developed spinal walking (Table 2 and Table 3). There was no statistically significant difference observed in comparing these findings (*p* = 0.129).

Regarding the duration of deep pain loss, it was observed that 56.6% (17/30) of dogs in PG and 28.6% (6/21) in CG lost deep pain sensation for up to 24 h, 16.6% (5/30) in PG and 38.1% (8/21) in CG between 25 and 48 h, 20% (6/30) in PG and 28.6% (6/21) in CG between 49 and 72 h, and 6.6% (2/30) in PG and 4.8% (1/21) in CG between 73 and 96 h. There was no statistically significant difference observed when comparing these results with functional recovery (*p* = 0.87).

## 4. Discussion

The therapeutic modalities employed in this study closely align with those documented in previous research [7,27,28], with a notable emphasis on the use of an aquatic treadmill (post-suture removal), as well as NMES and PROM exercises initiated early in paraplegic dogs [28]. These interventions aim to enhance weight-bearing capacity, increase afferent pathways (input), boost reflex arcs, optimize posture, and maximize recovery [29]. Importantly, no difficulties were encountered in executing any of the therapeutic modalities employed, with no need for exercise substitution. This was likely achieved through patient adaptation and conditioning in each exercise session (one to two sessions), as well as the addition of resources such as balls, treats, presence of the owner, and other individual toys [30].

The frequency of physiotherapy sessions varies across studies, ranging from intensive daily sessions over 90 days [31], daily for 42 days [5,7], daily for 30 days [3], to between 1 and 122 sessions of physiotherapy [32]. These variations in session number and duration of physiotherapy have led to differing outcomes regarding the influence of physiotherapy on the rate and time of return to ambulation.

The methodology adopted in this study aimed to rigorously evaluate the contribution of post-operative physiotherapy to functional neurological recovery in paraplegic dogs with LN due to thoracolumbar IVDE. In this study, efforts were made to establish a control group (without physiotherapy), including dogs with the same neurological condition (paraplegia without DPP) and using the same physiotherapeutic modalities. However, isolating the specific effects of physiotherapy for comparison with a control group presents significant challenges.

In an attempt to establish a minimum timeframe to assess the effects of physiotherapy on functional recovery in surgically treated paraplegic dogs without nociception due to IVDE, literature sources were consulted, but no specific references were found on this information. However, one study suggested a minimum period of 21 days of physiotherapy treatment for dogs with thoracolumbar IVDE to observe any influence of physiotherapy on recovery [33]. Nevertheless, it cannot be definitively stated that physiotherapy cannot contribute to the recovery of dogs who received treatment for less than 21 days, such as up to six physiotherapy sessions, considering two sessions per week. Therefore, the authors of this study opted to compare results based on the time it took dogs in the PG and CG to achieve satisfactory recovery within 21 days or beyond this period post-surgery.

Regarding functional neurological recovery, 26/51 dogs (51%) showed satisfactory recovery. Among these, 3/30 dogs (11.5%) in PG achieved satisfactory recovery within 21 days post-operatively, while 4/21 dogs (15.4%) in CG reached this condition within the same period. After 21 days post-operatively, it was observed that 10/30 dogs (38.5%) were in PG and 9/21 dogs (34.6%) were in CG. There was no statistically significant difference between these results in the two groups (*p* = 1.000).

The recovery rate in dogs with acute spinal cord injuries varies and depends on several factors, including the severity of the damage, the onset time of neurological signs, and the type of treatment employed [15,34,35]. Studies have reported different rates of motor recovery in dogs with severe injuries caused by IVDE, ranging from 30 to 75% [15,16,36,37]. An association between the time of onset of neurological signs and the recovery of paraplegic dogs with absent nociception was not observed in a study [15]. However, they emphasize the influence of the severity of spinal cord contusion injuries and the complexity in discussing functional recovery rates in dogs with different degrees of spinal cord injuries, resulting in a wide range of data disparities. In this study, the treatment type (surgery) and onset time of neurological signs (up to 96 h) were standardized. Regarding the severity of injuries, only paraplegic animals with LN were included.

The difficulty in demonstrating the impact of physiotherapy on the rate and time of ambulation recovery may be attributed to the inclusion of dogs with varying degrees of neurological dysfunction, lack of standardization of physiotherapy protocols, primarily the absence of a negative control group (without physiotherapy) [8,19], as evidenced in previous studies [7,28,31,32] and the divergence of definitions of satisfactory recovery.

In this study, we considered unconscious walking (spinal walking), that is, motor activity without sensory activity, as unsatisfactory recovery, because we aimed to evaluate the return of all spinal cord functions [12,38]. Not all studies of recovery after spinal cord injury interpret “satisfactory functional recovery” in this way [15]. This divergence in definition can lead to confusion in interpreting the results. If we considered spinal walking as satisfactory recovery, the rate of patients in this condition in PG would be 66.66% (20/30), yet it would still not differ from CG, at 61.9% (13/21).

It was observed that 22.6% (7/30) of dogs in PG and 9.53% (2/21) in CG developed spinal walking; however, physiotherapy did not appear to influence this outcome compared to CG (*p* = 0.129). Conversely, intensive physiotherapy in paraplegic dogs without nociception enabled the development of involuntary, unassisted walking in 59% of patients [39]. One study reported that 58.33% of paraplegic dogs (35/60) with LN due to IVDE or thoracolumbar fracture developed spinal walking after 125 to 320 sessions of physiotherapy (25 to 64 weeks), involving manual therapy (massage), electrostimulation, ultrasound therapy, laser therapy, and hydrotherapy [40].

The effectiveness of physiotherapy as part of post-operative treatment in dogs with IVDE has been extensively studied. Authors who sought this answer through prospective, randomized, or blinded studies did not observe differences in motor recovery rates when adding physiotherapeutic modalities after decompressive spinal cord surgery [7,28,41]. However, a study compared a group of dogs treated with postoperative physiotherapy to a negative control group and observed higher rates of motor recovery in animals undergoing physiotherapy [42]. However, upon analyzing the results and methodology, dogs with paraplegia and absent nociception who received only surgical treatment had a recovery rate of only 17%, below values reported in the literature. The lack of details on control group selection created the possibility of treatment bias.

Experimental studies in animals have demonstrated that intensive physiotherapy training can induce anatomical and physiological changes in the spinal cord, potentially leading to improvements in ambulation [43,44]. While human medicine has widely applied the results of these experimental studies, especially in applying physiotherapy to patients with paraplegia, the protocols and techniques used are not commonly used by physiatrists in veterinary medicine. For instance, robotic gait training, which has shown benefits in humans with severe spinal cord injuries, cannot be easily adapted for animals due to high costs and challenges associated with inducing specific voluntary movements through commands [45,46]. Other gait training sessions without the use of robots, with the physiotherapist modulating the movements of the patients’ limbs on an aquatic treadmill, terrestrial treadmill, or various types of ground surfaces, are also conducted and yield motor recovery [47]. Moreover, the type of physiotherapy known to improve locomotor outcomes in experimental models and currently used in people with severe spinal cord injuries is highly intensive locomotor training. The activities described in these locomotor training protocols significantly exceed the intensity of the therapeutic activities commonly implemented in veterinary medicine [8], which traditionally include the techniques employed in this study.

A controlled retrospective study compared conventional physiotherapy with intensive neurorehabilitation. In the intensive rehabilitation group, gait training in paraplegics was conducted similarly to the methods applied in humans mentioned above, with gait modulated by the physiotherapist on terrestrial and aquatic treadmills, with elevated intensity and frequency. In the group of dogs with LN, the authors observed a significantly higher rate of return to ambulation in animals treated in the intensive protocol group compared to the control group, at 55.8% and 33.2%, respectively [28]. However, not all animals that regained ambulation recovered limb sensitivity and developed spinal gait, although this recovery was considered a satisfactory response in the study’s methodology. This result contrasts with the approach of the present study, which aimed to evaluate the return of complete spinal function. Furthermore, it is possible to conclude that intensive training, similar to that applied in humans and experimental studies, may bring benefits to the return of unconscious ambulation. However, doubts remain as to whether this type of rehabilitation can lead to complete spinal cord recovery, that is, with conscious walking, preserving both motor and sensory functions.

Although not used in this research, other therapeutic modalities have been employed in the post-operative care of dogs with IVDE as photobiomodulation [48,49], pulsed electromagnetic field (PEMF) and electroacupuncture.

Low-level laser therapy or photobiomodulation has been a reported therapy for various injuries, including spinal cord injury. In an unblinded, unrandomized prospective study of non-ambulatory paraparetic or paraplegic dogs with IVDE, with or without intact pain perception at enrolment and treated surgically, laser therapy was applied post-operatively (for 5 days or until independent ambulation was achieved) was compared to dogs that did not receive additional therapy. The reported time to achieve independent ambulation was shorter in the laser therapy group (3.5 days) compared to the untreated control dogs (14 days) [50]. However, the characteristics of the laser employed were not detailed, making it difficult to try to replicate results. In contrast, a blinded, randomized prospective study evaluating post-operative laser therapy with or without physical rehabilitation in non-ambulatory dogs undergoing surgery for IVDE revealed no difference in recovery [41]. Importantly, both studies included a relatively small number of dogs in each treatment group, including few with severe injury, did not incorporate pre-study sample size calculations, and only looked at short-term outcome variables. No adverse events attributable to laser therapy were reported.

Pulsed electromagnetic field therapy has also been used for post-operative analgesia and the recovery of locomotion in dogs. The study found that PEMF reduced incision-associated pain in dogs post-surgery, decreased the extent of spinal cord injury, and improved proprioceptive positioning [51]. The authors also reported a possible neurologic benefit based on measuring injury severity using plasma GFAP concentration and the recovery of proprioceptive placing. Oscillating electrical field therapy, which is suggested to enhance axonal regrowth and improve functional recovery, has been applied to spinal cord-injured animals [52]. In paraplegic deep-pain-negative dogs secondary to IVDE treated surgically, oscillating electrical field therapy was delivered post-operatively via electrodes sutured to the edges of the laminectomy site and attached to an implantable device. Treatment was administered for a variable number of weeks post-operatively and the device and therapy were well-tolerated. Dogs treated with the PEMF had improved neurologic outcomes at 6 weeks and 6 months after surgery compared with sham-treated dogs [52,53].

Electroacupuncture has also proven effective in the recovery of ambulation and pain control. In one retrospective and two prospective case series of dogs with thoracolumbar IVDE of variable severity, electroacupuncture (administered 1–3 ×/week for 1–6 months between studies) was reported to be more effective than decompressive surgery alone for regaining ambulation [54], and was associated with a shorter time to walking and a greater proportion of dogs becoming ambulatory compared to medical management alone [38,55]. There was no significant difference in recovery among deep-pain-negative dogs managed medically with or without electroacupuncture [38]. Study limitations for the prospective studies included lack of blinding or randomization, use of historical controls and small sample size within each neurologic grade [38,55]. There is equivocal evidence that electroacupuncture decreases the severity and duration of post-operative pain in dogs with IVDE [38,56]. Electroacupuncture has also been combined with stem cell transplantation in a small group of dogs chronically (>3 months), deep-pain-negative following acute IVDE [57]. This pilot study showed that these interventions were feasible and safe, but the case numbers in each treatment group were small.

A limitation of this research is its retrospective nature and the non-random distribution of the dogs in the studied groups. For the formation of the negative control group, dogs that did not receive physiotherapy treatment were included due to the distance and impracticality of accessing a specialized center, as they lived in remote areas. Although this may introduce some bias into the study, we believe that it did not interfere with the observed results, and to ensure the validity of GC, the owners were interviewed by telephone to confirm that the dogs were confined to moving only inside the house, were not encouraged to walk or engage in other active exercises, and did not undergo any physiotherapy techniques. This helped ensure the establishment of a true negative control group.

Despite these limitations, it was possible to form a negative control group. When comparing the recovery rate within 21 days and beyond that period between the PG and CG groups, no significant difference was found, suggesting that there was no favoring of functional neurological recovery for a particular group. Additionally, another potential benefit would be regarding the duration of PDP and the recovery rate, which did not differ statistically between the studied groups. Other limitations included the small sample size of dogs due to strict inclusion criteria aimed at reducing variables that could influence the recovery rate.

Despite the mentioned limitations, the formation of a negative control group in the present study enabled a thorough analysis of the role of post-operative physiotherapy in the functional recovery of paraplegic dogs with LN due to IVDE. Thus far, it was not shown to influence functional neurological recovery.

## 5. Conclusions

Based on the proposed methodology and the results obtained in the study, it was concluded that the physiotherapeutic protocol applied to paraplegic dogs with loss of nociception due to extrusion of the thoracolumbar intervertebral disc does not seem to influence functional neurological recovery compared to the group without physiotherapy.

## Figures and Tables

**Figure 1 animals-14-02648-f001:**
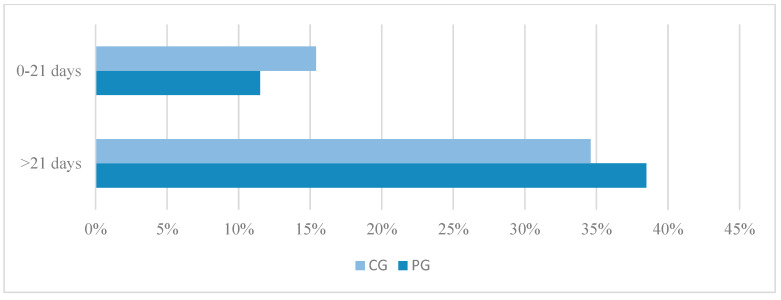
The functional recovery outcome of paraplegic dogs with loss of nociception due to intervertebral disc extrusion, surgically treated with (PG) and without (CG) physiotherapy before and after 21 days postoperatively.

**Figure 2 animals-14-02648-f002:**
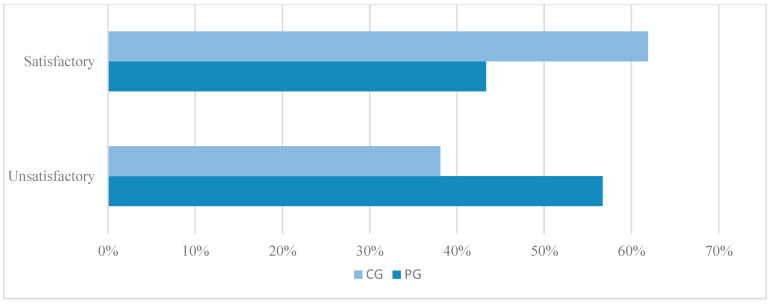
Functional recovery outcome of paraplegic dogs with loss of nociception due to intervertebral disc extrusion, surgically treated with (PG) and without (CG) postoperative physiotherapy.

**Table 1 animals-14-02648-t001:** Distribution of breed, age, sex, compression site, time of deep pain perception loss, functional recovery, and time to functional recovery in paraplegic dogs without nociception due to intervertebral disc extrusion submitted for decompressive surgery with or without postoperative physiotherapy.

	PG	CG	Total
Dogs	30 (58.8%)	21 (41.2%)	51 (100%)
Breed *n* (%)	Dachshund 19 (63.3%) SRD 8 (26.7%) Poodle 1 (3.3%) Yorkshire 1 (3.3%) Pekingese 1 (3.3%)	Dachshund 15 (71.4%) SRD 2 (9.5%) Poodle 1 (4.7%) Pug 1 (4.7%) Cocker 1 (4.7%)	Dachshund 34 (66.6%) SRD 10 (19.6%) Poodle 2 (3.9%) Yorshire 1 (1.9%) Pekingese 1 (1.9%) Pug 1 (1.9%) Cocker 1 (1.9%)
Age (years)	Min 2	Min 3	Min 2
	Max 9	Max 10	Max 10
	Mean 5.18 ± 1.83	Mean 5.69 ± 1.52	Mean 5.56 ± 2.05
Sex	Female 15 (50%)	Female 11 (52.4%)	Female 26 (50.9%)
	Male 15 (50%)	Male 10 (47.6%)	Male 25 (49.1%)
Compression site	T11–T12 = 1 (3.3%)	T11–T12 = 3 (14.3%)	T11–T12 = 4 (7.8%)
	T12–T13 = 10 (33.3%)	T12–T13 = 2 (9.5%)	T12–T13 = 12 (23.5%)
	T13–L1 = 15 (50%)	T13–L1 = 10 (47.6%)	T13–L1 = 25 (49.1%)
	L1–L2 = 1 (3.3%)	L1–L2 = 4 (19.1%)	L1–L2 = 5 (9.8%)
	L2–L3 = 3 (10%)	L2–L3 = 2 (6.5%)	L2–L3 = 5 (9.8%)
Duration nociception loss (hours)	<24 *n* = 10 (33.3%)	<24 *n* = 3 (14.4%)	<24 *n* = 13 (25.5%)
	25–48 *n* = 12 (40%)	24–48 *n* = 12 (57.2%)	24–48 *n* = 24 (47.1%)
	49–7 *n* = 6 (20%)	49–72 *n* = 6 (28.6%)	49–72 *n* = 12 (23.5%)
	73–96 *n* = 2 (6.7%)	73–96 *n* = 1 (4.8%)	73–96 *n* = 3 (8.9%)
Functional recovery	Satisfactory: *n* = 13 (43.3%)	Satisfactory: *n* = 13 (61.9%)	Satisfactory: *n* = 26 (51%)
	Unsatisfactory: *n* = 17 (56.7%)	Unsatisfactory: *n* = 8 (38.1%)	Unsatisfactory: *n* = 25 (49%)
	Total: *n* = 30	Total: *n* = 21	Total: *n* = 51
Time functional recovery (days)	<21: *n* = 3 (23.1%)	<21: *n* = 4 (31%)	<21: *n* = 7 (13.7%)
	>21: *n* = 10 (76.9%)	>21: *n* = 9 (69%)	>21: *n* = 19 (37.2%)
	Total: *n* = 13	Total: *n* = 13	Total: *n* = 26

PG: physiotherapy group; CG: control group; *n*: number of dogs; SRD: mixed breed.

**Table 2 animals-14-02648-t002:** Representation of breed, sex, age, duration of deep pain perception loss, compression site, functional recovery, return time, and presence and onset of spinal gait in surgically treated paraplegic dogs with loss of nociception due to intervertebral disc extrusion, not submitted for physiotherapy.

CG	Breed, Sex, Age (Years)	DDPP (Hours)	SC	FR	TRF (Days)	SW	OSW (Months)
1	SRD, M, 5	24	L2–L3	S	<21		
2	Dachshund, M, 5	24	L1–L2	S	<21		
3	Dachshund, M, 5	24	T13–L1	S	<21		
4	Cocker, F, 5	51	T11–T12	S	>21		
5	Dachshund, F, 7	48	T11–T12	S	>21		
6	Dachshund, F, 6	96	T13–L1	S	>21		
7	Dachshund, F, 7	57	L2–L3	S	>21		
8	Poodle, F, 4	56	T13–L1	S	>21		
9	SRD, M, 5	64	T11–T12	S	>21		
10	Dachshund, M, 7	48	T13–L1	S	>21		
11	Pug, F, 8	67	T13–L1	S	>21		
12	Dachshund, M, 5	72	T13–L1	I	-	Y	3
13	Dachshund, F, 5	12	L1–L2	I	-		
14	Dachshund, M, 7	12	T13–L1	I	-		
15	Dachshund, F, 3	12	L1–L2	I	-		
16	Dachshund, M, 5	48	L1–L2	S	<21		
17	Dachshund, M, 5	48	T13–L1	S	>21		
18	Dachshund, M, 10	48	T12–T13	I	-		
19	Dachshund, F, 7	35	T12–T13	I	-	Y	5
20	Dachshund, M, 4	48	T12–T13	I		Y	3
21	Dachshund, F, 6	48	T12–T13	I	-		

CG: control group; M: male; F: female; SRD: mixed breed; DDPP: duration deep pain sensation loss; SC: site compression; FR: functional recovery; TFR: time functional recovery; SW: spinal walking; OSW: onset spinal walking; S: satisfactory; I: unsatisfactory; Y: yes.

**Table 3 animals-14-02648-t003:** Representation of breed, sex, age, duration of deep pain sensation loss, site of compression, physiotherapy protocol, number of physiotherapy sessions, functional recovery, return time, and presence and onset of spinal walking in paraplegic dogs with loss of nociception due to intervertebral disc extrusion treated surgically and submitted for postoperative physiotherapy.

PG	Breed, Sex, Age (Years)	DDPP (Hours)	SC	PP	NPS	FR	TFR (Days)	SW	OSW (Months)
1	Dachshund, M, 5	<24	T12–T13	P1 + P2	24	I	-		
2	Dachshund, F, 5	<24	T12–T13	P1 + P2	24	I	-		
3	Dachshund, M, 5	<24	T13–L1	P1 + P2 + P3	22	S	>21		
4	Pekingese, M, 4	72	L2–L3	P1 + P2 + P3	60	S	>21		
5	Dachshund, M, 8	72	T11–T12	P1 + P2 + P3	14	S	>21		
6	SRD, M, 5	48	T13–L1	P1 + P2	22	I	-	Y	4
7	Dachshund, M, 3	<24	T12–T13	P1 + P2 + P3	21	I	-		
8	Dachshund, F, 6	<24	T12–T13	P1	10	I	-		
9	SRD, F, 4	72	T12–T13	P1 + P2	18	I	-	Y	3
10	Dachshund, M, 5	72	T13–L1	P1 + P2	22	I	-		
11	Dachshund, M, 6	<24	T13–L1	P1 + P2 + P3	18	S	>21		
12	Dachshund, M, 6	24	T12–T13	P1	10	I	-	Y	4
13	Dachshund, F, 4	24	T13–L1	P1	26	I	-		
14	Poodle, M, 6	96	T13–L1	P1	10	I	-		
15	Dachshund, F, 9	24	T3–L3	P1 + P2 + P3	12	S	>21		
16	SRD, F, 2	<24	T13–L1	P1 + P2 + P3	9	S	>21		
17	SRD, M, 9	<24	T13–L1	P1 + P2 + P3	7	S	<21		
18	SRD, F, 5	<24	T12–T13	P1 + P2 + P3	6	S	<21		
19	SRD, M, 6	96	T12–T13	P1 + P2	17	I	-	Y	3
20	SRD, F, 7	<24	T13–L1	P1 + P2 + P3	24	S	>21		
21	Dachshund, F, 4	24	L2–L3	P1 + P2 + P3	10	S	>21		
22	SRD, F, 8	72	T11–T12	P1 + P2	22	I	-	Y	3
23	Dachshund, F, 3	24	T11–T12	P1 + P2 + P3	10	S	>21		
24	Dachshund, F, 4	48	T13–L1	P1	18	I	-	Y	2
25	Dachshund, M, 10	48	T13–L1	P1 + P2	22	I	-		
26	Dachshund, M, 11	72	L1–L2	P1 + P2	20	I	-	Y	3
27	Dachshund, F, 4	24	T12–T13	P1	19	I	-		
28	Dachshund, M, 6	48	L2–L3	P1 + P2 + P3	22	I	-		
29	Yorkshire, F, 4	24	T13–L1	P1 + P2 + P3	24	S	>21		
30	Dachshund, F, 4	48	T13–L1	P1 + P2 + P3	6	S	<21		

PG: physiotherapy group; M: male; F: female; SRD: mixed breed; DDPP: duration deep pain sensation loss; SC: site compression; PP: physiotherapy protocol; NPS: number physiotherapy sessions; FR: functional recovery; TFR: time functional recovery; SW: spinal walking; OSW: onset spinal walking. Y: yes; P1: effleurage massage, passive range motion (PROM), flexor reflex stimulation, and neuromuscular electrical stimulation (NMES); P2: aquatic treadmill + assisted walking with body slings; P3: walking on mattress, walking with obstacles and use of the circular proprioceptive platform.

## Data Availability

The data presented in this study are available upon request from the corresponding author.

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
