# Peer review of "Physiotherapy in the Recovery of Paraplegic Dogs without Nociception Due to Thoracolumbar Intervertebral Disc Extrusion Treated Surgically"

_animals, 2024, doi:10.3390/ani14182648_

Round 1

Reviewer 1 Report

Comments and Suggestions for Authors

Thank you for proposing an interesting protocol for animal physiotherapy research.  I do have significant concerns with regards to the conclusion and definition of satisfactory outcome, especially when this appears to be based on return of nociception alone?

Line 44: Interesting as I do not believe that the same can be said in the USA. Anecdotally, we see more orthopaedic conditions in veterinary rehabilitation than neurologic. Understanding that this is cited in an article about neuro rehabilitation, is this relative to number of visits or number of patients?

Line 46: Challenging in the human rehabilitation literature as well, physiotherapy and its techniques are not always well defined.  Can we be more specific about what techniques are used, at what variables (load, frequency, intensity) and might this have affected results of this study?

Line 114: Can you further define/describe “sliding massage” as I am unfamiliar with that technique?

Line 114-116: Can you further elaborate on these “modalities” which might better be described as “techniques”?  Specifically, what variables were used? 

Line 119-123: Can you further describe or include a photo of the reflex stimulation and ROM and stretching activities?  Where was the animal handled and stimulated? Sensory input is integral to SCI rehabilitation in human patients and, I suspect, in animal as well.

Line 132-136: Was an assistant in the UWT facilitating or patterning movements of the limbs? Was the patient weight bearing when a sling was used for assisted walks? On what type of surface—pavement, tile, dirt, grass, etc?

Line 150: Why is this an unsatisfactory result? The pet is walking but doesn’t have nociception. This is misleading with regards to your conclusion as the walking (even without nociception) can be/might be considered functional return. See the article by Lewis, Jeffery, and Olby  which suggests that functional return can occur without return of deep pain or nociception.

“Ambulation in Dogs With Absent Pain Perception After Acute Thoracolumbar Spinal Cord Injury

Melissa J. Lewis1*, Nick D. Jeffery2, Natasha J. Olby3 and the Canine Spinal Cord Injury Consortium (CANSORT-SCI)” https://www.researchgate.net/publication/343881285_Ambulation_in_Dogs_With_Absent_Pain_Perception_After_Acute_Thoracolumbar_Spinal_Cord_Injury

Line 183: Length of physiothereapy was 18.5 weeks, however, outcomes were only assessed through 120 days? Would/did this affect the outcome for some in the PG group?

Line 217: It is unclear to me whether the spinal walking dogs were considered satisfactory or unsatisfactory outcomes?

Line 221: “had it” meaning had deep pain or had deep pain loss?  This is confusing with regards to wording.  The statistics examined each of the classification groups (specifically, comparied PG to CG in the “up to 24 hours” group)? By my impression of the % examined, it looks like there was a difference?  Can you explain in more detail how the numbers were examined for statistical significance?

Line 230: MPA exercises was not defined earlier in the text.  Can you elaborate?

Line 249: I appreciate the difficulty in establishing a protocol as well as defining a control group.  But control was not crate rest or restriction but was ad lib or client chosen interventions (not professional)? What were clients doing in the CG?

Line 257: “In this study, it was observed that 22.6% (7/30) of dogs in the 257 Physiotherapy Group (PG) and 9.53% (2/21) in the Control Group (CG) developed spinal 258 walking; however, physiotherapy did not appear to influence this outcome compared to 259 the control group.”  What was the influence?  Why the difference?  I’m confused.

Line 310: Yes, I agree, there is confusion in the interpretation of satisfactory result.  Whose interpretation? As a therapist, I consider satisfactory to be functional, independent gait. However, you are referring to satisfactory result as return of complete neurological function including sensation? I think this is a very important point and should be clarified in the abstract, introduction, and results/conclusion/discussion.

Line 319: Good point, but in human rehabilitation/physiotherapy, robotic training actually has a reduced outcome compared to body weight support gait training in an activity based therapy approach. This BWSGT and ABT can be used in veterinary rehabilitation. 

Line 322: I think this statement is incorrect, suggesting that robotic training is the only human physiotherapy strategy for SCI? There are many others and many that can be adapted to animal patients.

Line 325: Yes, intensity is high and that is necessary for successful functional recovery. Are you suggesting that animal physio/rehab therapists increase the intensity of their therapy for improved outcomes?

Line 337: This statement is unclear. How is this a concern? Please elaborate/clarify this statement.

Line 343: I’m concerned that the control group was not exactly controlled. Control would mean restricted? Or activity ad lib?

Line 348: I disagree with how this statement is made. Unsatisfactory outcome is not the same as functional recovery.  These dogs did not recover nociception. Many did recover functional mobility via spinal walking or volitional walking.

Further notes:

-        I would suggest a more thorough evaluation/review of the human SCI rehab literature. It goes beyond robotic locomotor therapy.

-        Also, in the bibliography, “espinal” should be corrected to “spinal”.

Author Response

Dear reviewers. We appreciate the suggestions made to make the text more comprehensible. All suggestions have been accepted and are highlighted in yellow.

Line 44: Interesting as I do not believe that the same can be said in the USA. Anecdotally, we see more orthopaedic conditions in veterinary rehabilitation than neurologic. Understanding that this is cited in an article about neuro rehabilitation, is this relative to number of visits or number of patients?

In Brazil, rehabilitation centers also predominantly treat neurological cases due to the high demand from patients with neurological issues. The sentence has been corrected to reflect the predominance of neurological cases in veterinary rehabilitation. (Line 55-56)

Line 46: Challenging in the human rehabilitation literature as well, physiotherapy and its techniques are not always well defined.  Can we be more specific about what techniques are used, at what variables (load, frequency, intensity) and might this have affected results of this study?

Corrected in the text. (line 124-157)

Line 114: Can you further define/describe “sliding massage” as I am unfamiliar with that technique?

Corrected in the text. (line 132-137)

Line 114-116: Can you further elaborate on these “modalities” which might better be described as “techniques”?  Specifically, what variables were used?

Corrected in the text. (line 132-203)

Line 119-123: Can you further describe or include a photo of the reflex stimulation and ROM and stretching activities?  Where was the animal handled and stimulated? Sensory input is integral to SCI rehabilitation in human patients and, I suspect, in animal as well.

Corrected in the text. If you still deem it necessary, we can send the images. (Line 137-152)

Line 132-136: Was an assistant in the UWT facilitating or patterning movements of the limbs? Was the patient weight bearing when a sling was used for assisted walks? On what type of surface—pavement, tile, dirt, grass, etc?

Corrected in the text. (line 167-180)

Line 150: Why is this an unsatisfactory result? The pet is walking but doesn’t have nociception. This is misleading with regards to your conclusion as the walking (even without nociception) can be/might be considered functional return. See the article by Lewis, Jeffery, and Olby which suggests that functional return can occur without return of deep pain or nociception.

We have carefully reviewed the article and made the necessary corrections to the text (Lines 197-203). In our study, we aim to assess the complete functional recovery of the spinal cord in dogs affected by IVDD. Functional recovery of the spinal cord resulting from thoracolumbar injury is defined by motor and sensory recovery below the level of the lesion, specifically in the pelvic limbs. Once sensory recovery is achieved, a truly functional recovery of the spinal cord, where all pathways and tracts are functioning, has not been possible. OLBY; LEVINE; HARRIS; MUNANA; SKEEN; SHARP, 2003 (Long-term functional outcome of dogs with severe injuries of the thoracolumbar spinal cord: 87 cases (1996–2001)) did not consider spinal walking as successful recovery. In the article you mentioned, the authors use the term "functional ambulation."

Given this ambiguity in interpretations, the entire text has been revised, and this explanation of satisfactory functional recovery has been clarified throughout. If the reviewer still considers spinal walking to be a satisfactory recovery, the authors of the article are willing to make the change.

Line 183: Length of physiothereapy was 18.5 weeks, however, outcomes were only assessed through 120 days? Would/did this affect the outcome for some in the PG group?

Corrected to 8.85 weeks. Inserted in the text. Line 235-236

Line 217: It is unclear to me whether the spinal walking dogs were considered satisfactory or unsatisfactory outcomes?

"Dogs with spinal walking were considered as an unsuccessful, i.e., unsatisfactory outcome, due to incomplete spinal cord recovery (lack of sensory function recovery - deep pain). We considered spinal walking as unsatisfactory based on the article (Long-term functional outcome of dogs with severe injuries of the thoracolumbar spinal cord: 87 cases (1996–2001)) which did not consider spinal walking as a successful outcome."

Line 221: “had it” meaning had deep pain or had deep pain loss?  This is confusing with regards to wording.  The statistics examined each of the classification groups (specifically, comparied PG to CG in the “up to 24 hours” group)? By my impression of the % examined, it looks like there was a difference?  Can you explain in more detail how the numbers were examined for statistical significance?

“Had it” means “they had lost the deep pain”. I skimmed the text for better understanding. (Line 273-278).

This statistical analysis was conducted to observe if there was a difference in the duration of deep pain loss between the PG and CG groups. A contingency table analysis (duration of deep pain loss vs. PG and CG) was created, and the chi-square test was applied. The animals were grouped based on the duration of deep pain loss: <24h, 24-48h, 49-72h, and 73-96h for both PG and CG. Comparisons were made as follows: PG <24h vs. CG <24h; PG 24-48h vs. CG 24-48h; PG 49-72h vs. CG 49-72h; PG 73-96h vs. CG 73-96h.

Although the percentages appear different, the analysis revealed that this difference is not statistically significant.

Line 230: MPA exercises was not defined earlier in the text.  Can you elaborate?

Corrected to PROM (it was in Portuguese). Line 282

Line 249: I appreciate the difficulty in establishing a protocol as well as defining a control group.  But control was not crate rest or restriction but was ad lib or client chosen interventions (not professional)? What were clients doing in the CG?

"We agree with the lack of explanation regarding the management of animals in the control group. It was recommended by the veterinarian that the patient could only move indoors, without encouragement for walks or other active exercises. These explanations have been added to the text. (Lines 102-103)"

Line 257: “In this study, it was observed that 22.6% (7/30) of dogs in the 257 Physiotherapy Group (PG) and 9.53% (2/21) in the Control Group (CG) developed spinal 258 walking; however, physiotherapy did not appear to influence this outcome compared to 259 the control group.”  What was the influence?  Why the difference?  I’m confused.

"In this sentence, we attempted to explain that although more dogs in the physiotherapy group developed spinal walking, statistically comparing with the control group showed no difference. Based on this result, it suggests that physiotherapy did not appear to influence the development of spinal walking under the conditions of our study when compared to the group without physiotherapy. (Lines 337-352)"

Line 310: Yes, I agree, there is confusion in the interpretation of satisfactory result.  Whose interpretation? As a therapist, I consider satisfactory to be functional, independent gait. However, you are referring to satisfactory result as return of complete neurological function including sensation? I think this is a very important point and should be clarified in the abstract, introduction, and results/conclusion/discussion.

"We agree with your statement. Therefore, the entire text was revised to better understand this definition. If deemed necessary, we can include cases of spinal walking as satisfactory outcomes. Some articles, for example (Long-term functional outcome of dogs with severe injuries of the thoracolumbar spinal cord: 87 cases (1996–2001), Functional outcome in dogs undergoing hemilaminectomy for thoracolumbar disc extrusion but without nociception > 96 h: A prospective study), have included cases of return to ambulation without deep pain as another classification of outcomes."

Line 319: Good point, but in human rehabilitation/physiotherapy, robotic training actually has a reduced outcome compared to body weight support gait training in an activity based therapy approach. This BWSGT and ABT can be used in veterinary rehabilitation.

We agree. Added a discussion of these techniques to the text (line 363-379).

Line 322: I think this statement is incorrect, suggesting that robotic training is the only human physiotherapy strategy for SCI? There are many others and many that can be adapted to animal patients.

I didn't write it to be the only one, but it was mentioned as a technique. In view of your statement, a discussion about other techniques has been added to the text. (Lines 380-394).

Line 325: Yes, intensity is high and that is necessary for successful functional recovery. Are you suggesting that animal physio/rehab therapists increase the intensity of their therapy for improved outcomes?

Yes. We suggest that studies with high-intensity protocols be conducted to better assess the effects of this type of rehabilitation in paraplegic patients. So far, there have been no studies of intensive neurorehabilitation comparing with a negative control group (without physiotherapy). That's why we suggest it.

Line 337: This statement is unclear. How is this a concern? Please elaborate/clarify this statement.

Corrected in the text (lines 442-455).

Line 343: I’m concerned that the control group was not exactly controlled. Control would mean restricted? Or activity ad lib?

We agree with the lack of explanation regarding the management of animals in the control group. It was recommended by the veterinarian that the patient could only move indoors, without encouragement for walks or other active exercises. These explanations have been added to the text. (Lines 102-103).

Line 348: I disagree with how this statement is made. Unsatisfactory outcome is not the same as functional recovery.  These dogs did not recover nociception. Many did recover functional mobility via spinal walking or volitional walking.

Corrected in the text. Lines 462-465.

Further notes:

-        I would suggest a more thorough evaluation/review of the human SCI rehab literature. It goes beyond robotic locomotor therapy.

Dear reviewer, the human literature was consulted and some observations were inserted into the text.

-        Also, in the bibliography, “espinal” should be corrected to “spinal”.

Corrected in text

Reviewer 2 Report

Comments and Suggestions for Authors

Thank you for the opportunity to review your manuscript on the role of physiotherapy in the recovery of paraplegic dogs lacking nociception due to thoracolumbar intervertebral disc extrusion treated surgically. The manuscript is well written, easy to follow and understand. I have several questions and thoughts that I would like addressed: 

  1. 1. You state that this is a retrospective study. How were dogs divided into receiving physiotherapy vs not? 

  1. 2. Since no significant difference was found between the groups, might this be a factor of group size? Was a power analysis performed? 

  1. 3. You discuss the role of early gait retraining in humans and the difficulty in translating to dogs (e.g. impracticality of robotic gait trainers). What are your thoughts on earlier introduction to gait patterning in a physiotherapy program for this population of dogs? Your protocol involved bicycling in lateral recumbency starting from P1 while P2 initiated underwater treadmill or sling assisted walking only once motor was appreciable.

  2. What are your thoughts on performing bicycling/gait patterning of a plegic patient with sling support in a standing position, or on a land treadmill or underwater treadmill during P1? These more closely mimic normal walking and stimulate the rhythmic pattern generators required for gait compared to bicycling of one limb at a time in a laterally recumbent position. These techniques have been described as benefiting gait training of dogs who develop spinal walking with persistent absent nociception in other studies, so I wonder if earlier gait patterning introduction could be beneficial more immediately post-op in the recovery phase of plegic deep pain negative dogs. 

  1. 4. Other modalities that have been looked at for supporting post-surgical recovery from IVDD include photobiomodulation, acupuncture/electroacupuncture, and pulsed electromagnetic field therapy, all with variable results. I think it would be worthwhile to add to the discussion brief references to these other modalities as they are some of the most commonly utilized (at least in the United States) for post-operative hemilaminectomy.

  2. 5. The difficulty from a rehab practitioner standpoint with many similar studies to yours is differences in rehab protocols and how this may affect recovery response. I think it is important to make it clear in the conclusion that this particular protocol used in the study demonstrated no significant difference in functional recovery rate compared to the control group. This ties in to the prior comments about earlier gait retraining methods and the role of other modalities in this population of dogs and whether these could potentially influence the recovery rate.

Author Response

Dear reviewers. We appreciate the suggestions made to make the text more comprehensible. All suggestions have been accepted and are highlighted in yellow.

  1. You state that this is a retrospective study. How were dogs divided into receiving physiotherapy vs not?

The dogs in the physiotherapy group received physiotherapy treatment at the Hospital's Physiotherapy Department, while the control group consisted of dogs whose owners lived far from rehabilitation centers, making it impractical for them to attend physiotherapy sessions. To ensure this information, owners were contacted by phone and asked about not undergoing any rehabilitation techniques after surgery, and all confirmed that they did not.

  1. Since no significant difference was found between the groups, might this be a factor of group size? Was a power analysis performed?

The small number of animals could be a limiting factor, and perhaps with a larger sample size, different results could be observed. However, the inclusion criteria were quite rigorous, resulting in a small sample size. Based on power analysis, the sample size should ideally be 55 animals per group.

  1. You discuss the role of early gait retraining in humans and the difficulty in translating to dogs (e.g. impracticality of robotic gait trainers). What are your thoughts on earlier introduction to gait patterning in a physiotherapy program for this population of dogs? Your protocol involved bicycling in lateral recumbency starting from P1 while P2 initiated underwater treadmill or sling assisted walking only once motor was appreciable.

What are your thoughts on performing bicycling/gait patterning of a plegic patient with sling support in a standing position, or on a land treadmill or underwater treadmill during P1? These more closely mimic normal walking and stimulate the rhythmic pattern generators required for gait compared to bicycling of one limb at a time in a laterally recumbent position. These techniques have been described as benefiting gait training of dogs who develop spinal walking with persistent absent nociception in other studies, so I wonder if earlier gait patterning introduction could be beneficial more immediately post-op in the recovery phase of plegic deep pain negative dogs.

It seems like you're discussing the use of intensive neurorehabilitation protocols in canine patients recovering from severe acute intervertebral disc extrusion. The study by Ângela Martins and colleagues (2021) utilized such a protocol, focusing on paraplegic dogs with and without deep pain perception. Although they observed successful outcomes in terms of return to walking even in dogs without deep pain perception, the authors concluded that the contribution of intensive neurorehabilitation to this recovery couldn't be fully assessed. Their findings suggest that the success of walking ability can potentially be improved, particularly concerning the time taken for recovery.

  1. Other modalities that have been looked at for supporting post-surgical recovery from IVDD include photobiomodulation, acupuncture/electroacupuncture, and pulsed electromagnetic field therapy, all with variable results. I think it would be worthwhile to add to the discussion brief references to these other modalities as they are some of the most commonly utilized (at least in the United States) for post-operative hemilaminectomy.

Included in the text (line 395-441).

  1. The difficulty from a rehab practitioner standpoint with many similar studies to yours is differences in rehab protocols and how this may affect recovery response. I think it is important to make it clear in the conclusion that this particular protocol used in the study demonstrated no significant difference in functional recovery rate compared to the control group. This ties in to the prior comments about earlier gait retraining methods and the role of other modalities in this population of dogs and whether these could potentially influence the recovery rate.

We agree. Clarified in conclusion. Lines 462-465.